# Measurement of Serum Testosterone in Nondiabetic Young Obese Men: Comparison of Direct Immunoassay to Liquid Chromatography-Tandem Mass Spectrometry

**DOI:** 10.3390/biom10121697

**Published:** 2020-12-19

**Authors:** Ana Martínez-Escribano, Julia Maroto-García, Maximiliano Ruiz-Galdón, Rocío Barrios-Rodríguez, Juan J. Álvarez-Millán, Pablo Cabezas-Sánchez, Isaac Plaza-Andrades, María Molina-Vega, Francisco J. Tinahones, María Isabel Queipo-Ortuño, José Carlos Fernández-García

**Affiliations:** 1Department of Surgery, Biochemistry and Immunology, Málaga University, 29010 Málaga, Spain; anamtezescribano@gmail.com (A.M.-E.); juliamarotogarcia@gmail.com (J.M.-G.); mruizg@uma.es (M.R.-G.); 2Departmento de Medicina Preventiva y Salud Pública, Universidad de Granada, 18011 Granada, Spain; rbarrios@ugr.es; 3Consortium for Biomedical Research in Epidemiology and Public Health (CIBERESP), 28029 Madrid, Spain; 4Instituto de Investigación Biosanitaria (ibs.GRANADA), 18014 Granada, Spain; 5Consulting Químico Sanitario (CQS Lab), 28521 Madrid, Spain; jamillan@cqslab.com (J.J.Á.-M.); pcabezas@cqslab.com (P.C.-S.); 6Department of Medical Oncology, Virgen de la Victoria and Regional University Hospitals-IBIMA, UMA-CIMES, 29010 Málaga, Spain; isaacplazaandrade@gmail.com; 7Department of Endocrinology and Nutrition, Virgen de la Victoria University Hospital, 29010 Malaga, Spain; molinavegamaria@gmail.com (M.M.-V.); josecarlosfdezgarcia@hotmail.com (J.C.F.-G.); 8Laboratorio de Investigación, Instituto de Investigación Biomédica de Málaga (IBIMA), 29010 Málaga, Spain; 9Centro de Investigación Biomédica en Red de la Fisiopatología de la Obesidad y Nutrición (CIBEROBN), Instituto de Salud Carlos III (ISCIII), 28029 Málaga, Spain; 10Department of Endocrinology and Nutrition, Hospital Regional Universitario de Málaga, 29010 Malaga, Spain

**Keywords:** testosterone, obesity, immunoassay, chromatography

## Abstract

Hypoandrogenemia, a frequent finding in men with obesity, is defined by low concentrations of serum testosterone. Although immunoassay (IA) is the most used method for the determination of this steroid in clinical practice, liquid chromatography-mass spectrometry (LC-MS/MS) is considered a more reliable method. In this study, we aimed to compare IA versus LC-MS/MS measurement for the diagnosis of hypoandrogenemia in a cohort of 273 nondiabetic young obese men. Mean total testosterone (TT) levels were 3.20 ± 1.24 ng/mL for IA and 3.78 ± 1.4 ng/mL for LC-MS/MS. 53.7% and 26.3% of patients were classified as presenting hypoandrogenemia with IA and LC-MS/MS, respectively. Considering LC-MS/MS as the reference method, sensitivity and specificity of IA were 91.4% (95% CI 82.3–96.8) and 61.1% (95% CI 54.0–67.8), respectively. IA presented an AUC of 0.879 (95% CI 0.83–0.928). Multivariate regression analysis indicated that sex hormone-binding globulin (SHBG) concentrations (*p* = 0.002) and insulin resistance (*p* = 0.008) were factors associated with discrepant IA values. In conclusion, the determination of TT by IA in nondiabetic young men with obesity yields lower concentrations of TT than LC-MS/MS, resulting in an equivocal increased diagnosis of hypoandrogenemia, which could lead to inaccurate diagnosis and unnecessary treatment.

## 1. Introduction

Hypoandrogenemia, a term used to denominate the finding of subnormal testosterone concentrations in men, independently of associated clinical symptoms or signs of decreased testosterone levels [1], is described as a frequent comorbidity in men. Therefore, in the general population, hypoandrogenemia affects between 2.1% and 12.8% of adult men [2].

Although there are many conditions associated with decreased concentrations of testosterone, including diabetes mellitus [3], aging [4], and cardiovascular disease (CVD) [2], obesity is considered to be the major cause of hypoandrogenemia in men [5,6]. In this regard, a recent study of our group has reported that hypoandrogenemia affects more than 25% of young obese males without diabetes mellitus or CVD [7].

Undoubtedly, reliable and accurate measurement of testosterone is required to diagnose patients properly where hypoandrogenemia is suspected. In this context, the Endocrine Society has elaborated different consensus and standardization documents recommending clinicians to determine testosterone levels with accurate and precise technology rather than with simple and economic methods [8,9].

Currently, immunoassay (IA) is the most used method for the determination of testosterone given the low requested sample volume, and because it represents a simple, fast, and inexpensive analytical method [9,10]. Unfortunately, a number of studies have reflected several disadvantages in testosterone direct IA determination. Specifically, inaccuracy has often been described, particularly in conditions with low testosterone levels, such as in females and children [11,12]. In fact, Taieb et al. [13] found that none of the ten IA (non-isotopic and isotopic IA) tested were reliable enough for the investigation of the very low (0.17 nmol/L) and low (1.7 nmol/L) testosterone concentrations expected in sera from children and women.

Moreover, due to the low antibody specificity of this technology, over- or underestimation may be observed with different assays [13,14], and, finally, not all IA run with the same quality development [15]. In addition, several studies have shown that testosterone IA have method-specific bias relative to the gold standard mass spectrometry-based reference methods. In fact, Sikaris et al. [16] found significant differences between several automated IA and the independent gas chromatography–mass spectrometry (GC/MS) reference method for serum testosterone assays in blood samples from healthy eugonadal young men, as well as substantial discrepancies between methods in the reference intervals, particularly in its lower limit. Wang et al. examined serum testosterone levels in samples from eugonadal and hypogonadal males by different automated and manual IA methods and compared their results with measurements performed by liquid chromatography-tandem mass spectrometry (LC-MS/MS). Over 60% of the samples with testosterone levels within the adult male range measured by most automated and manual IA were within +/−20% of those reported by LC-MS/MS. However, for testosterone values less than 5.2 nmol/liter, the values were neither analytically nor clinically useful [17]. In addition, Rosner et al. (appointed by the Council of The Endocrine Society) compared studies from multiple laboratories using different methods such as IA and mass spectrometry to measure plasma testosterone in samples from adult normal female and hypogonadal men or androgenized women. They found problems with the variability and the lack of standards for accuracy between the methods. In addition, the authors propose that the best prospect for a gold standard lies in extraction and LC-MS/MS or MS-MS in which the chemical structure of the molecule measured is identified [9].

As a result, LC-MS/MS has been considered a more suitable method for steroid hormone measurement, given its higher precision and specificity compared to IA [18]. However, LC-MS/MS also has some disadvantages, basically its more complicated and expensive performance. Therefore, LC-MS/MS is not frequently used in laboratory practice [19,20].

In particular, it has been reported that, in obesity, the determination of serum total testosterone (TT) by IA can be both biased towards lower or higher TT values, in comparison with LC-MS/MS. This can be due to the different assays used, but also due to the characteristics of the study population [21]. Specifically, as previously highlighted, the inclusion of elderly subjects or patients with diabetes mellitus or established CVD can strongly influence the concentrations of TT [2,3,4].

Therefore, in this study, we aimed to compare direct testosterone IA versus LC-MS/MS measurement for the diagnosis of hypoandrogenemia in a cohort of young obese men without diabetes or CVD.

## 2. Materials and Methods

### 2.1. Study Subjects

In this study, we included young (<50 years) adult men with obesity [body mass index (BMI) ≥30 kg/m^2^]. These subjects were randomly selected from six primary care centers located in the metropolitan area of Malaga, Spain; in each primary care center, five primary care physicians were randomly selected, and each primary care physician consecutively invited 10 young adult men with obesity to participate in the study. Men with diabetes mellitus, men under treatment with androgens and subjects with hepatic, renal impairment, CVD or cancer were excluded from participating in this study.

### 2.2. Study Samples

Study participants were advised to fast with an effect from 10 p.m. Participants completed a structured interview to obtain demographic and clinical data, including height, waist circumference (WC) and blood pressure. Fasting blood samples were collected before 10:00 a.m. and were allowed to clot at room temperature for 30 min prior to centrifugation (3000 rpm for 10 min). Serum was immediately removed and distributed in aliquots. All samples were collected in the Outpatient Clinic of the Endocrinology Department of the Virgen de la Victoria Hospital, and were immediately transported to the Medical Laboratory Department where they were bar-coded and blinded to patient identification. We used the homeostatic model assessment of insulin resistance (HOMA-IR) to determine the status of insulin resistance [22]. In non-diabetic men, 1.85 was considered as the cut-off value of insulin resistance [23]. Serum insulin levels were measured by immunoassay using an ADVIA Centaur autoanalyzer (Siemens, Erlangen, Germany). Weight and body composition were obtained using the Tanita multi-frequency body composition analyzer MC-180MA (Tanita Corporation, Tokyo, Japan).

### 2.3. Testosterone Analysis

Testosterone assays were performed by IA and LC-MS/MS. All samples were measured similarly to other test samples run in each laboratory. No testosterone values below assay sensitivity were found. To convert conventional to SI units: TT (ng/mL) × 3.467 (nmol/L) was used.

#### 2.3.1. Immunoassay

Testosterone IA run on the fully automated analyzer Advia Centaur (Siemens Healthcare, Erlangen, Germany) TT assay is a competitive immunoassay based on chemiluminescence technology. Sample testosterone competes with acridinium ester labelled testosterone for testosterone polyclonal rabbit antibodies bound to paramagnetic solid-phase particles. Luminescence is initiated by the addition of an acid and base reagent after magnetic separation and washing of the particles. All measurements were performed according to the manufacturer’s instructions. A master curve is provided by the manufacturer for each lot of reagents. Calibrator E, levels H and L (CAL E for Advia Centaur, Siemens), was used to generate the assay curve. Immunoassay Plus Control levels 1, 2 and 3 (Biorad Laboratories) were used with each run and to calculate inter- and intra-assay precision.

Detection limit studies were carried out in accordance with the CLSI EP17-A2 guideline. For IA, the limit of blank (LoB) was determined from 20 replicates of a zero material (the manufacturer provided analyte-free diluent). Limit of detection (LoD) was determined from the limit of blank and using the following equation: LoD = LoB + cβSDs where SDs is the estimated standard deviation of the sample distribution at a low level and cβ is derived from the 95th percentile of the standard Gaussian distribution. The LoQ pools were prepared by diluting patient sera. The LoQ is the lowest concentration that can be detected with ≤20% CV.

Finally, the coefficient of variation (CV) intra-assay was calculated after 20 repetitions of three different levels of commercially available quality control material in one day. CV inter-assay was obtained from the data of one month.

#### 2.3.2. LC-MS/MS

TT determination was carried out using MassChrom Steroids kit (Chromsystems, Munich, Germany) and high performance LC-MS/MS triple quadrupole equipment (Model 6460, Agilent Technologies, Santa Clara, CA, USA). The applied conditions for sample treatment and MS/MS determination were the indicated by the kit with slight modifications.

The calibrators were human serum sample based. The material was lyophilized and contained six levels of 0.05, 0.250, 0.982, 2.94, 5.82, 11.6 ng/mL, and a blank. Quality control (QC) samples were prepared from different stock solutions of human serum samples at three levels (0.202, 1.49, and 8.0 ng/mL serum concentration). Working calibrations and QC were reconstituted with 3 mL of distilled water in each vial and were incubated for 15 min at room temperature. The vials were gently swirled to dissolve the contents until homogeneity. Two hundred microliters of calibrators, QC, and sample, along with 50 µL of the internal standard (testosterone-d_3_), were subjected to a process of reversed phase 96-well solid-phase extraction prior to injection into the HPLC system. The eluates were evaporated to dryness by means of a nitrogen stream at 45 °C and reconstituted in 100 µL of initial mobile phase.

To carry out the analysis, 20 µL of processed sample were injected into the chromatographic system and were separated onto an HPLC column (Chromsystems, Gräfelfing, Germany). Two MRM transitions for testosterone were measured to check for interference (289.2–97 *m*/*z* and 289.2–109 *m*/*z*) and one transition for testosterone-d_3_ (292.2–97 *m*/*z*).

The gradient used for total testosterone quantification was 70% of phase A (aqueous phase with methanol) and 30% of phase B (water/acetonitrile) with a flow of 0.8 mL/min during the first minute of the analysis. Then, the percentage of phase B increased from 30% to 44% and continued until 3.50 min where the flow was lowered to 0.4 mL/min. This flow was maintained until min 7, where the percentage of phase B increased to 50% and a flow of 0.6 mL/min. In min 8.61, the flow rose to 0.8 mL/min and the percentage of phase B to 60%, reaching 100% and a flow of 1 mL/min in min 10.81. From min 11.51, they returned to the initial conditions until min 13. Finally, 2 min were left under these conditions to re-equilibrate the column until next injection.

The LC-MS/MS system was controlled with the Agilent MassHunter Workstation software version B.06.00 (Agilent Technologies, Santa Clara, CA, USA). For peak integration and quantitative calculation, the Agilent MassHunter Quantitative Analysis software version B.06.00 was used.

The LC-MS/MS was validated in serum samples by determining the linearity, precision, recovery, interference, limits of detection (LoD), and quantification (LoQ). The linear range of the present method was determined using a 6-point calibration curve (testosterone, 0.054–11.6 ng/mL) measured in four replicates and linearity was evaluated using a linear regression method. Correlation coefficient (R^2^) ≥ 0.999 was obtained during the method validation procedure. A calibration curve and three replicates at each QC were prepared on the same day to calculate the analytical concentrations (intra-assay variation). This procedure was carried out on ten different days (inter-assay variation), which allowed the evaluation of the stability of the method. To verify the trueness (recovery), three QC samples were prepared from different stock solutions of human serum samples at different levels (0.202, 1.49, and 8.0 ng/mL serum concentration). The recovery was calculated as [(final concentration-initial concentration)/added concentration] × 100%. Elimination of interferences in the measurement system can be achieved using different MRM channels for the same compound. Multiple reactions monitoring also enables application of the isotopically labelled internal standards (ISTD), which improves the accuracy. The LoQ was determined using five replicates as the lowest concentration, which generated a signal/noise ratio of 10 and CV <20% and the LoD represents the absolute limit of detection that produced a signal/noise ratio ≥3.

To ensure the quality of the LC-MS/MS results, external controls (serum samples) were used as a reference. These quality controls were obtained from the German Society for Clinical Chemistry and Laboratory Medicine (www.dgklrfb.de), belonging to the program HM1 (Survey for the determination of testosterone by LC-MS/MS). The goal of this external quality assessment is to demonstrate the competence through participation in inter-laboratory trials. This material was processed and analyzed at the same time as the study samples; the results obtained did not differ by more than 10% with regard to their certified value, therefore verifying the accuracy of our method.

### 2.4. Definition of Hypoandrogenemia

We used testosterone values from the LC-MS/MS assay as the reference for diagnosing hypoandrogenemia, since LC-MS/MS assays generally offer higher concentrations of specificity, sensitivity, and precision (especially in the low range) than do most immunoassays and also testosterone reference ranges are mostly based on LC-MS assays [8].

Hypoandrogenemia was defined as calculated FT values lower than 70 pg/mL [24]. FT was estimated from TT, sex hormone-binding globulin (SHBG) and albumin by using Vermuelen’s formula [25]. SHBG determination was performed by electrochemiluminescence immunoassay (Elecsys SHBG, Roche, Basel, Switzerland), with reference ranges of 15–50 nmol/L.

### 2.5. Ethical Approval

This study was reviewed and approved by the Ethics Committee of Virgen de la Victoria Clinical University Hospital, and was conducted according to the principles of the Declaration of Helsinki. The participants (who were all volunteers) provided signed consent after being fully informed of the study goal and its characteristics.

### 2.6. Statistical Analysis

Statistical analysis was performed using IBM SPSS Statistics 23.0 (IBM corporation, Armonk, NY, USA). Descriptive statistics were used to summarize empirical distributions of testosterone measurements from each assay. Passing Bablok regression analysis was performed to evaluate concordance between methods. Correlation coefficients (R) were calculated. Systematic and proportional bias were derived from the intercept (a) and slope (b), respectively, of the regression equation (Y = a + bX). Bland–Altman plots were used to assess the relative mean difference and corresponding 95% limit of agreement (±1.96*SD differences).

The clinical accuracy of the IA was analyzed by using Receiver Operator Characteristic (ROC) plots. The ROC area under the curve (AUC) was calculated using LC-MS/MS data as the gold standard. Sensitivity was defined as the proportion of patients correctly identified as having hypoandrogenemia, as initially diagnosed using LC-MS/MS in serum samples. Specificity was defined as the proportion of study participants accurately identified as negative for hypoandrogenemia.

In addition, a parsimonious multivariate logistic regression model was constructed to evaluate the factors associated with the disagreement of the testosterone results between IA and LC-MS/MS (defined by biases that differ >20% from the mean bias between IA and LC-MS/MS), taking into account multicollinearity (through the variance inflation factor). *p*-values <0.05 were considered as statistically significant.

## 3. Results

### 3.1. Cohort Descriptive Characteristics

Characteristics of the analysis cohort are shown in Table 1. Briefly, we included 273 men with a mean age of 37.2 ± 7.8 years, mean WC of 124.2 ± 15.3 cm, and mean BMI of 39.0 ± 6.8 kg/m^2^. Mean fasting glucose levels were 93.1 ± 10.0 mg/dl and mean HbA1c was 5.4 ± 0.4%. Testosterone IA values were lower than those obtained by LC-MS/MS; mean TT levels were 3.20 ± 1.24 ng/mL for IA and 3.78 ± 1.4 ng/mL for LC-MS/MS. The range of values went from 1.11 to 8.67 ng/mL with IA and 1.35 to 10.76 ng/mL with LC-MS/MS.

### 3.2. IA and LC-MS/MS Assessment

During IA assessment, the lower limit of detection (LoD) was 0.033 ng/mL; functional sensitivity was 0.062 ng/mL (the value provided by the manufacturer was 0.07 ng/mL). Intra-assay coefficients of variation (CV) were 2.6, 4.7, and 3.4% for mean concentrations of 0.98, 4.1 and 8.89 ng/mL. Inter-assay CV were 7.24, 6.69, and 7.57% respectively. Accuracy was 106.1%. Data are listed in Table 2.

Regarding LC-MS/MS, the lower LoD was 0.024 ng/mL, the inter-assay CV was 8.43% at 0.2 ng/mL, 2.64% at 1.49 ng/mL, and 2.64% at 8.08 ng/mL, and the intra-assay CV was 2.09%, 3.67%, and 1.64% at the mentioned concentrations, respectively. Accuracy was 103.7% and recovery was 97%. Data are listed in Table 2.

### 3.3. Correlation and Agreement between IA and LC-MS/MS

After Passing–Bablok regression analyses, the results obtained were slope 1.1603 (95% CI: 1.0735 to 1.2529) and intercept 0.0533 (95% CI: −0.2109 to 0.2876). The coefficient of correlation was 0.825. The estimated intercept included the 0, so there was no evidence for systematic bias between both assays. In addition, the slope did not include the 1, suggesting that the proportional difference between both methods may be different (Figure 1). Bland–Altman bias plot was used in the agreement analysis. As shown in Figure 2, IA underestimated testosterone values with a mean bias of −19.9 [95% limits of agreement (±1.96 SD); 20.7 to −60.5]. Only the mean values of 16.1% and 33.4% of the samples were within 5% and 10% of the target LC-MS/MS values, respectively. In addition, 58.6% of the mean values were found within ±20%.

### 3.4. Impact of Assay Performance on Clinical Assessment

According to the IA results, 143 of 273 patients were classified as presenting hypoandrogenemia (53.7%), while the proportion with the LC-MS/MS method was 70 out of 273 (26.3%). Considering LC-MS/MS as the reference method and fixing the threshold in FT < 70 pg/mL, sensitivity (% of true positives) and specificity (% of true negatives) of IA were 91.4% (95% CI 82.3–96.8) and 61.1% (95% CI 54.0–67.8), respectively. Positive predictive value (PPV) was 44% and negative predictive value (NPV) was 95%. Finally, the diagnostic performance of IA was also compared to LC-MS/MS by a ROC curve, presenting the IA method an AUC of 0.879 (95% CI 0.83–0.928).

### 3.5. Factors Involved in Testosterone Determination Disagreement

To evaluate the factors associated with the systematic IA bias in the determination of total testosterone, a multivariate model was constructed, considering as the dependent variable the biases between IA and LC_MS/MS determinations that differed >20% from the mean bias. In this multiple logistic regression analysis, the optimum model that best explained the presence of testosterone bias included SHBG concentrations (*p* = 0.002) and insulin resistance (*p* = 0.033) (Table 3).

## 4. Discussion

In this study, we have compared the analytical and clinical performance characteristic data between two different assays for TT in a cohort of nondiabetic young men with obesity. We have found that the IA method, compared with the LC-MS/MS method, underestimates TT values, resulting in the duplication of the diagnosis of hypoandrogenemia in this study population.

In our study, mean TT was lower in IA than in LC-MS/MS, as well as the lower and higher ranges. Moreover, the distribution of TT levels showed two different curves; data from IA were placed in lower values, while data from LC-MS/MS were displaced to the right.

Our results agree with those found by Montagna et al., who evaluated a non-obese cohort of European men, finding that median LC-MS/MS TT values were higher than IA values [26]. Nevertheless, in a study on adolescent girls (which likewise shows low testosterone levels) when chemiluminescence IA and LC-MS/MS were compared, median values were lower in LC-MS/MS [27].

In our study, the limit of detection (LoD) was, as expected, slightly lower in LC-MS/MS than IA, and the limit of quantification (LoQ) was similar in both assays. IA LoQ confirmed the manufacturer’s claim of 0.07 ng/mL. An LC-MS/MS LoQ claim was not provided by the manufacturer. This was far from the 0.035 ng/mL reported as Centaur functional sensibility [14] but in concordance with others who also reported LoD < 0.08 ng/mL [28].

Intraassay imprecision varied among the assays. Low level replications had the best CV in IA, whereas, in LC-MS/MS, it showed the higher CV. At level QC 2, which corresponds to a testosterone concentration within the reference interval for men, CV was similar in both assays. This demonstrates the fact that the assays are optimized for the mean TT concentrations for men. Accuracy was also similar in both assays (103% for LC-MS/MS and 106% for IA), while Centaur IA has been reported to have a significant greater tendency to over-recover testosterone (average of 168%) [14].

The Bland–Altman results presented a not so desirable mean bias of −19.9. In a recent study, a similar bias (−17%) was described for men, reporting a lower bias for women (−13.2%) [14]. In this study, a poorer correlation in boys and girls was obtained (−58.4 and −54.1, respectively), and lower TT values (<1.9 ng/mL) did not meet minimum performance criteria. These facts support the initial concerns regarding assay performance at lower testosterone concentrations, such as found in boys, girls and TT values < 1.9 ng/mL. Other authors have described different bias in four IA platforms, two of them presenting a mean bias over 20% [26]. In our study, the mean values of 58.6% of the samples were within ±20% of the target LC-MS/MS values. A similar percentage of IA values that fell outside the 20% of the LC-MS/MS values has already been described: 39.6% for radioimmunoanalysis and 48.5% and 50.4% for two different IA [17]. Moreover, the difference was notable in samples with low testosterone levels, a fact that is also shown when comparing radioimmunoanalysis to LC-MS/MS [29].

These differences in the assay performance may have a direct impact on the classification of the gonadal status of patients. Thus, given that FT levels under 70 pg/mL are considered below the 2.5th percentile in the reference American and European population [24], approximately 25% of subjects would have hypoandrogenemia based on the LC-MS/MS assay, but more than 50% of our population could have hypoandrogenemia based on IA assay. It is important to highlight that we did not use different cut-off values for hypoandrogenemia depending on whether the testosterone determination was done with IA or with LC-MS, and we used the LC-MS/MS values as the reference. The rationale for this was that no differentiated cut-offs values for both techniques are available and, in addition, most clinical guidelines use testosterone cut-off values derived from LC-MS/MS assays [8].

In a study based on the test of a serum sample whose TT value was 4.02 ng/mL, 15% of all laboratories reported values below 3 ng/mL (5th percentile TT reference levels), suggesting that patients could be misclassified as being androgen deficient depending on the laboratory where the testing was performed [30]. This might have a great impact on patients under testosterone replacement therapy, given that incorrect decisions about the efficacy of testosterone replacement therapy might be made depending on the laboratory where the testing is performed [28,30,31].

Consistent with the ROC curve, we established an IA sensitivity of 91.4% (82.3–96.8), and a specificity of 61.1% (54.0–67.8). In addition, it presented an AUC = 0.879, which could be considered fair, but not suitable for diagnostic purposes. Therefore, the poor specificity obtained would not allow clinicians to use IA testosterone results in the diagnostic approach of male patients with suspected hypoandrogenemia.

Finally, in our study, we found that the factor associated with discrepant IA values were SHBG concentrations and insulin resistance. These findings are in agreement with the fact that insulin resistance is a common finding in patients with obesity and with the close interrelation between obesity and SHBG homeostasis [7].

Our study has certain limitations but also some important strengths. We have studied a relatively small sample (although more elevated than in many previous studies), and, therefore, it should be validated in larger populations. In addition, we used calculated FT by means of Vermuelen’s formula; although this calculation has been proposed as an adequate alternative to equilibrium dialysis determination [8], some authors have highlighted that calculated FT could be inaccurate [32,33]. On the other hand, the strengths of our study lie in the careful design, including only young obese subjects without diabetes mellitus or CVD, excluding patients with other comorbidities and avoiding confounding factors that might interfere in TT measurement.

## 5. Conclusions

The determination of TT by IA in nondiabetic young men with obesity yields lower concentrations of TT than LC-MS/MS, resulting in equivocal increased diagnosis of hypoandrogenemia, which could lead to inaccurate diagnosis and unnecessary treatment.

## Figures and Tables

**Figure 1 biomolecules-10-01697-f001:**
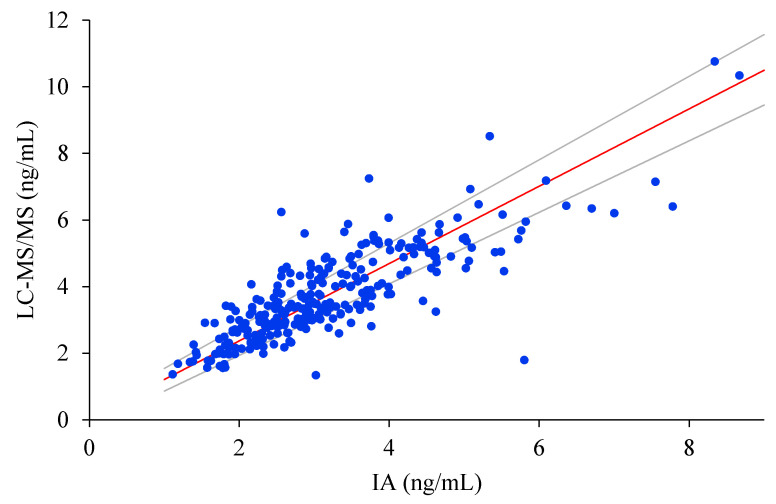
Passing–Bablok regression analysis for comparison between the LC-MS/MS method (*y*-axis) and the IA method (*x*-axis). Y = 0.0533 + 1.1603x (dark solid line).

**Figure 2 biomolecules-10-01697-f002:**
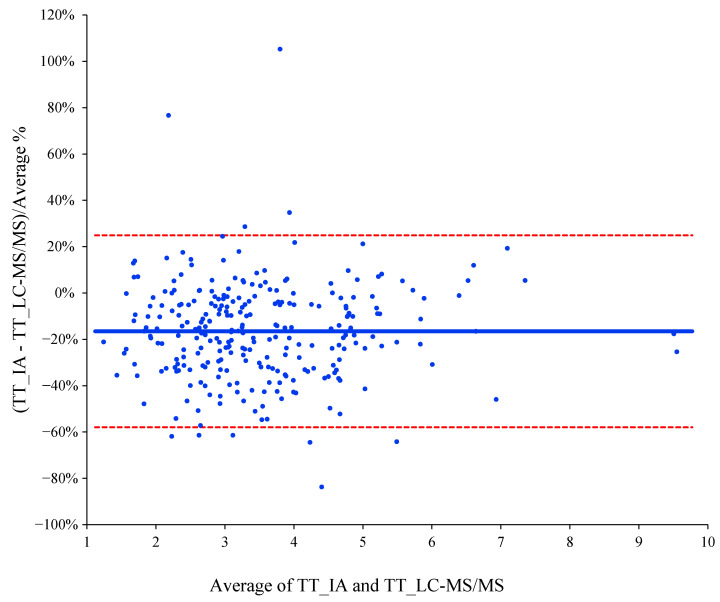
Bland–Altman bias plots for comparisons between the LC-MS/MS method and the IA. The solid line represents the relative mean difference between paired measurements; the dashed line indicates the upper and lower limits of agreement which approximate the mean measurement discrepancy ±1.96 standard deviations.

**Table 1 biomolecules-10-01697-t001:** Clinical characteristics of the study population (*n* = 273).

Characteristics	Mean ± SD	Median (p 50)	p 2.5	p 97.5
Age—years	37.2 ± 7.8	38	21.00	49.15
BMI—kg/m^2^	39.0 ± 6.8	37.63	30.55	56.73
WC—cm	124.2 ± 15.3	122	101.9	160
Fat mass—%	42.62 ± 14.79	39.6	23.14	78.57
Glucose—mg/dl	93.1 ± 10.0	91	78.00	122.3
Insulin—uIU/mL	20.1 ±15.4	16.93	6.44	51.42
HbA1c—%	5.4 ± 0.4	5.4	4.78	6.2
hs-CRP—mg/dl	3.45 ± 5.97	1.77	0.17	34.99
LH—mUI/mL	3.86 ± 2.11	3.39	1.31	8.14
TT (IA)—ng/mL	3.20 ± 1.24	2.95	1.43	6.41
TT (LC-MS/MS)—ng/mL	3.78 ± 1.40	3.46	1.67	6.96
SHBG—nmol/L	25.9 ± 12.7	23.5	9.90	55.32
FT (LC-MS/MS)—pg/mL	90.17 ± 30.9	86.6	42.03	169
FT (IA)—pg/mL	74.7 ± 26.9	69.1	32.56	143.9

Data are mean ± SD. SD: standard deviation; p: percentile; BMI, body mass index; WC, waist circumference; HbA1c, glycated hemoglobin; hs-CRP, high-sensitivity C-reactive protein; LH, luteinizing hormone; TT, total testosterone; IA: immunoassay; LC-MS/MS: liquid chromatography-mass spectrometry; SHBG, sex hormone-binding globulin; FT, free testosterone.

**Table 2 biomolecules-10-01697-t002:** Characteristics obtained in the evaluation of testosterone assay methods.

Assay	Mean ± SD	Range		LoD	LoQ	Imprecision			Accuracy
		Low	High			ng/mL	CV% Intraassay	CV% Interassay	%
IA	3.20 ± 1.24	1.11	8.67	0.033	0.062	0.98	2.6	7.24	106.1
						4.1	4.7	6.69	
						8.89	3.4	7.57	
LC-MS/MS	3.78 ± 1.4	1.35	10.76	0.024	0.073	0.2	2.09	8.43	103.7
						1.49	3.67	2.64	
						8.08	1.64	2.64	

Data shown in ng/mL. LoD: limit of detection; LoQ: limit of quantification.

**Table 3 biomolecules-10-01697-t003:** Factors associated with the disagreement of testosterone results between IA and LC-MS/MS (defined by biases between IA and LC_MS/MS determinations that differ> 20% from the mean bias) in nondiabetic young men with obesity: multivariate regression models.

		Multivariate	
	OR	95% CI	*p*
Age—years	0.985	0.949–1.022	0.432
HOMA-IR > 1.85	2.978	1.092–8.115	0.033
Fat mass—%	0.989	0.944–1.037	0.657
SHBG—nmol/L	1.042	1.015–1.070	0.002

IA: immunoassay; LC-MS/MS: Liquid chromatography-mass spectrometry; HOMA-IR, homeostatic model assessment of insulin resistance; SHBG: sex hormone-binding globulin. Logistic regression analysis: risk (odds ratio [OR]) of disagreement. Dependent variable: biases between IA and LC_MS/MS determinations that differ >20% from the mean bias.

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
