# Peer review of "Measurement of Serum Testosterone in Nondiabetic Young Obese Men: Comparison of Direct Immunoassay to Liquid Chromatography-Tandem Mass Spectrometry"

_biomolecules, 2020, doi:10.3390/biom10121697_

Round 1

Reviewer 1 Report

The study evaluates the performance of two methods for the testosterone determination in the diagnosis of hypogonadism in obese men.

The topic is actually more complex than how it is depicted in the article.

The number of hypogonadisms detected by the IA methods seems really very high.  However, although there are not reported the percentiles of the concentrations, from figure 1 we can roughly draw that the percentage of cases with insufficient concentrations of total testosterone found by the IA method could be around 13-30%, depending from the method reference interval and the different guidelines indications. Considering that in the obese subjects we could expect relatively low SHBG levels, the possible hypogonadism diagnosis could be further reduced. A possible explanation of the discrepancy could be the cut-off used for FT, that should be different for different methodologies.

Hypogonadism was defined by authors as FT <70 ng/mL, according the reference 19. However, the article of Matthews et al seems not related to testosterone, and I presume that the correct reference was 25, that found a threshold of 70 ng/mL referred at a LC/MS method. Then, probably the threshold could be not correct for the IA method. For example, the recent practice guideline of the Endocrine Society (Bhasin et al 2018) report a possible value of 64 pg/L, however referred, also in this case, to a LC/MS method (Wu et al 2010). Dittadi et al (2013) found, in a limited healthy subjects’ group, about 49 pg/mL, as a confirmation of the manufacturer’s reference interval of the IA method used in the study (32 pg/mL).

The article could be reconsidered if the classification of hypogonadism with IA will be re-evaluated according to a more appropriate threshold. Moreover, the diagnosis of hypogonadism is established also according some clinical features, as physical dysfunction and sexual symptoms. These characteristics should also be reported, considered in the classification and correlated to the testosterone levels.

Other remarks:

The bibliography should be verified (examples could be the error in the reference 26, and a possible problem of matching with the numbers reported in the text for the references 19/25).

Table 1 should also report the medians and the percentile distributions. There should be reported two FT data (obtained with IA and LC-MS)

The paragraph 3.2 should be developed. It should be described how the different LOD and CV were determined.

Author Response

Comment 1: The study evaluates the performance of two methods for the testosterone determination in the diagnosis of hypogonadism in obese men.

The topic is actually more complex than how it is depicted in the article.

The number of hypogonadisms detected by the IA methods seems really very high. However, although there are not reported the percentiles of the concentrations, from figure 1 we can roughly draw that the percentage of cases with insufficient concentrations of total testosterone found by the IA method could be around 13-30%, depending from the method reference interval and the different guidelines indications. 

Response 1: First of all, we thank the reviewer for his/her comments and for offering us a constructive review of our manuscript.

Regarding the prevalence of hypogonadism, it is indicated in the Results section, Subsection 3.4. Impact of assay performance on clinical assessment (Lines 270-271) and also in the abstract (Lines 35-36)

Lines 274-276:

- “According to the IA results, 143 of 273 patients were classified as presenting hypoandrogenemia (53.7%), while the proportion with the LC-MS/MS method was 70 out of 273 (26.3%).”

 Lines 35-36:

- “53.7% and 26.3% of patients were classified as presenting hypoandrogenemia with IA and LC-MS/MS, respectively”

Comment 2: Considering that in the obese subjects we could expect relatively low SHBG levels, the possible hypogonadism diagnosis could be further reduced. A possible explanation of the discrepancy could be the cut-off used for FT, that should be different for different methodologies. Hypogonadism was defined by authors as FT <70 ng/mL, according the reference 19. However, the article of Matthews et al seems not related to testosterone, and I presume that the correct reference was 25, that found a threshold of 70 ng/mL referred at a LC/MS method.

Response 2: Thanks for your remark.

As indicated by the reviewer, the correct reference was the number 25 (number 24 in the new version of the manuscript (Bhasin et al. Clin Endocrinol Metab. 2011 Aug;96(8):2430–9). Thanks for your careful review of the manuscript and sorry for this mistake.

 Comment 3: Then, probably the threshold could be not correct for the IA method. For example, the recent practice guideline of the Endocrine Society (Bhasin et al 2018) report a possible value of 64 pg/L, however referred, also in this case, to a LC/MS method (Wu et al 2010). Dittadi et al (2013) found, in a limited healthy subjects’ group, about 49 pg/mL, as a confirmation of the manufacturer’s reference interval of the IA method used in the study (32 pg/mL). The article could be reconsidered if the classification of hypogonadism with IA will be re-evaluated according to a more appropriate threshold.

Response 3: Thanks for your pertinent commentary.

As indicated by the reviewer, the testosterone threshold should be referred to the LC/MS method.

Indeed, in our study, the definition if hypoandrogenemia was based on the LC/MS method, not to the IA method. We acknowledge that this point maybe was not enough clear in the manuscript, therefore we have modified it:

 Lines 194-196:

 - “FT was estimated from TT (determined by LC-MS/MS), SHBG and albumin by using Vermuelen’s formula (25)”.

 On the other hand, regarding the threshold for hypoandrogenemia diagnosis, we based on a study by Bhasin et al (Bhasin S, et al. Reference ranges for testosterone in men generated using liquid chromatography tandem mass spectrometry in a community-based sample of healthy nonobese young men in the Framingham Heart Study and applied to three geographically distinct cohorts. J Clin Endocrinol Metab 2011;96:2430-9),where these authors established reference ranges for FT in a community-based sample of healthy nonobese young men. In this study, performed in 456 young men without obesity (mean BMI 25.5 ± 2.7 kg/m2), the 2.5th percentile value for FT was 70 pg/ml. Therefore, we used this cut-off for diagnosing hypoandrogenemia in our cohort (as our cohort was also conformed by young, nondiabetic, and free of cardiovascular disease men, but with obesity).

Comment 4: Moreover, the diagnosis of hypogonadism is established also according some clinical features, as physical dysfunction and sexual symptoms. These characteristics should also be reported, considered in the classification and correlated to the testosterone levels.

Response 4: Thanks for your remark.

Following the advice of reviewer 2, we have modified the term hypogonadism for hypoandrogenemia, which has been considered a more adequate term for this article, given that hypoandrogenemia is a more specific term coined to denominate the finding of subnormal testosterone concentrations in men, independently of associated clinical symptoms or signs of decreased testosterone levels (please, see comment 3 and response 3 from the second reviewer).

Comment 5: The bibliography should be verified (examples could be the error in the reference 26, and a possible problem of matching with the numbers reported in the text for the references 19/25).

Response 5: Sorry for the mistake. Bibliography has been verified along the manuscript.

Comment 6: Table 1 should also report the medians and the percentile distributions. There should be reported two FT data (obtained with IA and LC-MS).

Response 6: Thanks for your commentary. We have added the medians and the percentile distributions, as well as FT data for IA and LC-MS in the table 1.

Comment 7: The paragraph 3.2 should be developed. It should be described how the different LOD and CV were determined.

Response 7: Thanks for this remark.  We have added further information describing how LoD, LoQ and CV were determined. Also, we have included additional information for a better understanding of the techniques used in our study (M&M section of the revised manuscript): Lines 140-147: 

- “For IA, the limit of detection (LoD) was determined from 20 replicates of a zero material (the manufacturer provided analyte-free diluent). These responses were analyzed to determine the 2SD limit from the mean of the assay diluent and were used to obtain the minimum signal difference from zero in the calibration curve. The limit of quantification (LoQ) was determined from 20 replicates of the sample in which signal plus 2SD was different from zero. Finally, the coefficient of variation (CV) intra-assay was calculated after 20 repetitions of 3 different levels of commercially available quality control material in one day. CV inter-assay was obtained from the data of one month.”

Lines 166-189:

- “The LC-MS/MS was validated in serum samples by determining the linearity, precision, recovery, carryover, interference, limits of detection (LoD) and quantification (LoQ). The linear range of the present method was determined using a 6‑point calibration curve (testosterone, 0.054-11.6 ng/ml) measured in four replicates and linearity was evaluated using a linear regression method. Correlation coefficient (R2) ≥ 0.999 was obtained during the method validation procedure. A calibration curve and three replicates at each QC were prepared on the same day to calculate the analytical concentrations (intra-assay variation). This procedure was carried out on ten different days (inter-assay variation), which allowed the evaluation of the stability of the method. To verify the trueness (recovery) three QC samples were prepared from different stock solutions of human serum samples at different levels (0.202, 1.49 and 8.0 ng/ml serum concentration). The recovery was calculated as [(final concentration‑initial concentration)/added concentration]x100%. Carryover was done by injecting two blank samples subsequently after an upper limit of quantification sample in three independent runs. Moreover, at each work session, method blanks were used for monitoring interference. If traces of analytes were detected in method blanks, the values were subtracted from sample concentrations. The LoQ was determined using five replicates as the lowest concentration, which generated a signal/noise ratio of 10 and CV <20% and the LoD represents the absolute limit of detection that produced a signal/noise ratio ≥ 3.

To ensure the quality of the LC-MS/MS results, external controls (serum samples) were used as a reference. These quality controls were obtained from the German Society for Clinical Chemistry and Laboratory Medicine (www.dgklrfb.de), belonging to the program HM1 (Survey for the determination of testosterone by LC-MS/MS). The goal of this external quality assessment is to demonstrate the competence through participation in inter-laboratory trials. This material was processed and analyzed at the same time as the study samples; the results obtained did not differ by more than 10% with regard to their certified value, therefore verifying the accuracy of our method.”

Reviewer 2 Report

This is a solid assay method comparison of serum testosterone measurement in (obese) men. The best feature is its methodology using the best method comparison analyses, primarily based on Passing-Bablok and Bland-Altman approaches with good representation of reproducibility. However the study has other flaws which need further consideration

  1. A significant limitation of this manuscript is its very limited insight into the previous literature. This has long shown that testosterone immunoassays have method-specific bias relative to the gold standard mass spectrometry-based reference methods. They should cite the original studies showing - PMID 12881456, 16118337, 14764758 together with the US Endocrine Society’s Position Statement PMID 17090633. This should provide a significant caveat and limitation on interpretation of this manuscript in that it refers to only a single immunoassay and the findings reflect that immunoassay and not all other immunoassays. Indeed, this immunoassay is unusual that it gives lower values than LCMS but not without precedent and this could be discussed in this context of fitting in with prior literature. The analysis of reasons for IA discrepancy are largely pointless as they omit any reference to this particular immunoassay and deflect thinking misleadingly onto other variables - missing the forest for all the trees!
  2. The Bland-Altman approach gives an excellent overview of the apparent bias of this immunoassay. But more specific details are required to put it into proper clinical context. The manuscript should also specify what proportion of samples were within 5% and 10% of LCMS reference values and comment on this discrepancy.
  3. The definition of “hypogonadism” is unsatisfactory. Hypogonadism is a clinical term that refers to impaired function of the gonad, in this case the testes, as a clinical diagnosis of a pathological condition of the pituitary and testes, not a diagnosis based on laboratory numbers alone by any criterion. The study does not evaluate sperm output at all and neither infertility. It does refer to low serum testosterone which is not the same as “hypogonadism” or even testosterone (or androgen) deficiency if it is based on a single arbitrary cutpoint. This terminology should be corrected.
  4. The use of calculated FT is misguided and should be deleted as poitnless. These calculations based on Vermeulen are inaccurate (PMID 20346001, 20816687, 25240469, 19925815, 19225026) and clinically meaningless (PMID 28898980). Any reference to such calculations should specify that it is calculated and not masquerading as a real measurement with strong caveats on its meaningfulness.
  5. The title should delete the word “total” as it is serum testosterone that is being measured and any oblique reference to imaginary fractions of testosterone are beside the point of this study.

Author Response

This is a solid assay method comparison of serum testosterone measurement in (obese) men. The best feature is its methodology using the best method comparison analyses, primarily based on Passing-Bablok and Bland-Altman approaches with good representation of reproducibility. However the study has other flaws which need further consideration.

Comment 1: A significant limitation of this manuscript is its very limited insight into the previous literature. This has long shown that testosterone immunoassays have method-specific bias relative to the gold standard mass spectrometry-based reference methods. They should cite the original studies showing - PMID 12881456, 16118337, 14764758 together with the US Endocrine Society’s Position Statement PMID 17090633. This should provide a significant caveat and limitation on interpretation of this manuscript in that it refers to only a single immunoassay and the findings reflect that immunoassay and not all other immunoassays. Indeed, this immunoassay is unusual that it gives lower values than LCMS but not without precedent and this could be discussed in this context of fitting in with prior literature. The analysis of reasons for IA discrepancy are largely pointless as they omit any reference to this particular immunoassay and deflect thinking misleadingly onto other variables - missing the forest for all the trees!

Response 1: First of all, we would like to thank the reviewer for their useful comments and suggestions that undoubtedly have helped to improve our manuscript. As suggested, in this new version of the manuscript we have added a paragraph relating to the reviewer's suggestions:

- Lines 67-92 in the revised manuscript:

“Taieb et al. compared ten immunoassays (non-isotopic and isotopic immunoassays) and a non-immunochemical [isotope-dilution gas chromatography–mass spectrometry (GC/MS)] method to quantify low testosterone concentrations in healthy men, women and children. They found that none of the immunoassays tested were reliable enough for the investigation of the very low (0.17 nmol/L) and low (1.7 nmol/L) testosterone concentrations expected in sera from children and women. Moreover, the immunoassays underestimated testosterone concentrations in samples from men, giving mean results 12% below those obtained by the non-immunochemical method [15]. In addition, Sikaris et al. compared several automated immunoassays with GC/MS to measure serum total testosterone in blood samples from healthy eugonadal young men with verified normal reproductive function. They found significant differences between commercial automated immunoassays and the independent GC/MS reference method for serum testosterone assays, as well as substantial discrepancies between methods in the reference intervals, particularly in its lower limit [16]. In another study, Wang et al. examined six serum testosterone levels in samples from eugonadal and hypogonadal males by different automated immunoassay instruments and two manual immunoassay methods and compared their results with measurements performed by liquid chromatography-tandem (LC-MS/MS) mass spectrometry. Over 60% of the samples with testosterone levels within the adult male range measured by most automated and manual methods were within +/- 20% of those reported by LC-MS/MS. However, for testosterone values less than 5.2 nmol/liter, the values were neither analytically nor clinically useful [17]. Also, Rosner et al. (appointed by the Council of The Endocrine Society) compared studies from multiple laboratories using different methods such as IA and mass spectrometry to measure plasma testosterone in samples from adult normal female and hypogonadal men or androgenized women. They found problems with the variability and the lack of standards for accuracy between the methods. In addition, the authors propose that the best prospect for a gold standard lies in extraction and LC-MS/MS or MS-MS in which the chemical structure of the molecule measured is identified [9].”

Comment 2: The Bland-Altman approach gives an excellent overview of the apparent bias of this immunoassay. But more specific details are required to put it into proper clinical context. The manuscript should also specify what proportion of samples were within 5% and 10% of LCMS reference values and comment on this discrepancy.

Response 2: Thanks for your commentary. We have now added the proportion of samples within 5% and 10% of LCMS reference values in lines 262-264 and a comment about the discrepancy (Lines 326-331):

 Lines 258-260

- “Only the mean values of 16.1% and 33.4% of the samples were within 5% and 10% of the target LC-MS/MS values, respectively. 58.6% of the mean values were found within ±20%.”

Lines 321-326

- “In our study, the mean values of 58.6% of the samples were within ±20% of the target LC-MS/MS values. A similar percentage of IA values that fell outside the 20% of the LC-MS/MS values have been already described: 39.6% for radioimmunoanalysis and 48.5% and 50.4% for two different IA [17]. Moreover, the difference was notable in samples with low testosterone levels, a fact that is also shown when comparing radioimmunoanalysis to LC-MS/MS [29].”

Comment 3: The definition of “hypogonadism” is unsatisfactory. Hypogonadism is a clinical term that refers to impaired function of the gonad, in this case the testes, as a clinical diagnosis of a pathological condition of the pituitary and testes, not a diagnosis based on laboratory numbers alone by any criterion. The study does not evaluate sperm output at all and neither infertility. It does refer to low serum testosterone which is not the same as “hypogonadism” or even testosterone (or androgen) deficiency if it is based on a single arbitrary cutpoint. This terminology should be corrected.

Response 3: Thanks for this remark.

Although male hypogonadism has been described as a clinical syndrome that results from failure to produce physiological concentrations of testosterone, normal amounts of sperm, or both (Basaria S. Male hypogonadism. Lancet 2014;383(9924):1250-63), in agreement with the reviewer, we have changed the term “hypogonadism” for the term “hypoandrogenemia”, which is a more specific term coined to denominate the finding of subnormal testosterone concentrations in men, independently of associated clinical symptoms or signs of decreased testosterone levels (Liu PY, Iranmanesh A, Nehra AX, Keenan DM, Veldhuis JD. Mechanisms of hypoandrogenemia in healthy aging men. Endocrinol Metab Clin North Am 2005;34(4):935-55).

Thus, throughout the entire manuscript we have replaced hypogonadism for hypoandrogenemia, we have modified the first paragraph of the introduction, and we have changed reference 1:

Line 47-49:

-“Hypoandrogenemia, a term used to denominate the finding of subnormal testosterone concentrations in men, independently of associated clinical symptoms or signs of decreased testosterone levels [1], is described as a frequent comorbidity in men.”

Comment 4: The use of calculated FT is misguided and should be deleted as pointless. These calculations based on Vermeulen are inaccurate (PMID 20346001, 20816687, 25240469, 19925815, 19225026) and clinically meaningless (PMID 28898980). Any reference to such calculations should specify that it is calculated and not masquerading as a real measurement with strong caveats on its meaningfulness.

Response 4: We thank the reviewer for his/her observation. As indicated by the reviewer, in this study we have used calculated FT (Vermuelen´s formula) for diagnosing hypoandrogenemia in our patients with obesity.

Despite we agree with the reviewer that in some clinical situations calculated FT could be inaccurate, the latest Endocrine Society Clinical Practice Guideline (Testosterone Therapy in Men With Hypogonadism) indicates that “clinicians should determine free testosterone concentrations either directly from equilibrium dialysis assays or by calculations that use total testosterone, SHBG, and albumin concentrations” (Bhasin S, et al. Testosterone Therapy in Men With Hypogonadism: An Endocrine Society Clinical Practice Guideline. J Clin Endocrinol Metab 2018;103(5):1715-1744). A similar recommendation is given in the European Association of Urology Guidelines on Male Hypogonadism 2019 (available at https://uroweb.org/wp-content/uploads/EAU-Guidelines-on-Male-Hypogonadism-2019v2.pdf ): “For determination of FT levels, the calculation of FT with the help of the sex hormone binding globulin (SHBG) is recommended”.

Thus, on the recommendation of the reviewer, we have added in the limitations section the potential inaccuracy of calculated FT, and we have added two references given by the reviewer:

Line 349-352:

- “Also, we used calculated FT by means of Vermuelen’s formula; although this calculation has been proposed as an adequate alternative to equilibrium dialysis determination [8], some authors have highlighted that calculated FT could be inaccurate [32, 33].”

Comment 5: The title should delete the word “total” as it is serum testosterone that is being measured and any oblique reference to imaginary fractions of testosterone are beside the point of this study.​

Response 5: As indicated, we have deleted “total” in the title.

Reviewer 3 Report

The manuscript submitted by Ana Martínez-Escribano et al. attempted to establish a balanced method comparison between a conventional immunoassay (Advia Centaur, Siemens) and an LC-MS/MS-based method for the quantitation of serum testosterone. The results of 273 nondiabetic obese men were used for the diagnosis of hypogonadism, a prevalent disorder in this population, by both methods.

Major criticisms:

  1. Description the methods is incomplete, especially for the LC-MS/MS method. Information on sample preparation, chromatography and MS settings is essential for evaluation of an LC-MS/MS method. For example, extraction procedure, usage of internal standard used, type of chromatographic separation and MS settings could influence results greatly.
  2. Validation of the LC-MS/MS method is not described. LC-MS/MS methods are highly complex and require adequate validation of multiple parameters, such as imprecision, linearity, trueness, carryover, stability and interference. Only imprecision results are mentioned in the manuscript. However, it is difficult to interpret even these results without a proper explanation of the sample size and the experiment design. Also, especially in the context of this study, standardization with reference material (or at least harmonization with another properly validated LC-MS/MS assay) is necessary. Without this information, no conclusions can be stated on true differences between the results.
  3. Insufficient information is provided on the generation of the ROC data. Nothing is mentioned in the methods section as to how subjects are diagnosed by a reference method leading to the sensitivity and specificity data.
  4. Measurements by LC-MS/MS should be excluded from the multivariate model. LC-MS/MS measurements are an intrinsic part of the dependent variable you are trying to explain and have no added value here.
  5. Often the text is difficult to read due to incorrect punctuation, missing/wrong words or confusing paragraph structures. Before resubmission, I strongly recommend the authors to either improve the text themselves or send the text to an independent text editor.

Minor criticisms

  1. The authors refer to liquid chromatography tandem-mass spectrometry as LC_MS. Use either LC-MSMS or LC-MS/MS.
  2. The data in Figure 1 is largely already represented in Table 1. Therefore, Figure 1 has limited added value.
  3. The scales of Figure 2 are confusing. The x-axis starts at 1, while the y-axis starts at 0. This should be changed for the x-axis into 0.

Author Response

The manuscript submitted by Ana Martínez-Escribano et al. attempted to establish a balanced method comparison between a conventional immunoassay (Advia Centaur, Siemens) and an LC-MS/MS-based method for the quantitation of serum testosterone. The results of 273 nondiabetic obese men were used for the diagnosis of hypogonadism, a prevalent disorder in this population, by both methods.

Major criticisms:

Comment 1: Description the methods is incomplete, especially for the LC-MS/MS method. Information on sample preparation, chromatography and MS settings is essential for evaluation of an LC-MS/MS method. For example, extraction procedure, usage of internal standard used, type of chromatographic separation and MS settings could influence results greatly.

Response 1: We thank the reviewer for his/her observation.

Following the recommendation of the reviewer, information about sample preparation, chromatography and MS settings has been added to the revised manuscript (Lines 156-162):

- “Two hundred microliters of calibrators, QC, and sample, along with 50 ul of the internal standard (testosterone-d3), were pretreated on C-18 solid phase extraction. The eluates were evaporated to dryness by means of a nitrogen stream at 45°C and reconstituted. To carry out the analysis, 20 ml of processed sample were injected into the chromatographic system and were separated onto an HPLC column (Chromsystems, Gräfelfing, Germany). Two MRM transitions for testosterone were measured to check for interference (289.2­­─97 m/z and 289. 2­­─109 m/z) and one transition for testosterone-d3 (292.2­­─97 m/z).”

Comment 2: Validation of the LC-MS/MS method is not described. LC-MS/MS methods are highly complex and require adequate validation of multiple parameters, such as imprecision, linearity, trueness, carryover, stability and interference. Only imprecision results are mentioned in the manuscript. However, it is difficult to interpret even these results without a proper explanation of the sample size and the experiment design. Also, especially in the context of this study, standardization with reference material (or at least harmonization with another properly validated LC-MS/MS assay) is necessary. Without this information, no conclusions can be stated on true differences between the results.

Response 2: Thanks for your apt comment.

Following the recommendation of the reviewer, we have added further information about the validation of the LC-MS/MS method, including details about linearity, precision, recovery, carryover, interference, limits of detection, and limits of quantification:

Lines 166-182:

- “The LC-MS/MS was validated in serum samples by determining the linearity, precision, recovery, carryover, interference, limits of detection (LoD) and quantification (LoQ). The linear range of the present method was determined using a 6‑point calibration curve (testosterone, 0.054-11.6 ng/ml) measured in four replicates and linearity was evaluated using a linear regression method. Correlation coefficient (R2) ≥ 0.999 was obtained during the method validation procedure. A calibration curve and three replicates at each QC were prepared on the same day to calculate the analytical concentrations (intra-assay variation). This procedure was carried out on ten different days (inter-assay variation), which allowed the evaluation of the stability of the method. To verify the trueness (recovery) three QC samples were prepared from different stock solutions of human serum samples at different levels (0.202, 1.49 and 8.0 ng/ml serum concentration). The recovery was calculated as [(final concentration‑initial concentration)/added concentration]x100%. Carryover was done by injecting two blank samples subsequently after an upper limit of quantification sample in three independent runs. Moreover, at each work session, method blanks were used for monitoring interference. If traces of analytes were detected in method blanks, the values were subtracted from sample concentrations. The LoQ was determined using five replicates as the lowest concentration, which generated a signal/noise ratio of 10 and CV <20% and the LoD represents the absolute limit of detection that produced a signal/noise ratio ≥ 3.

Also, we have included further clarification about the design of the study (lines 108-111):

- “These subjects were randomly selected from six primary care centers located in the metropolitan area of Malaga, Spain; in each primary care center, five primary care physicians were randomly selected, and each primary care physician consecutively invited 10 young adult men with obesity to participate in the study.”

Finally, we have added a paragraph about the standardization with reference material in our LC-MS/MS assay (Lines 183-189).

- “To ensure the quality of the LC-MS/MS results, external controls (serum samples) were used as a reference. These quality controls were obtained from the German Society for Clinical Chemistry and Laboratory Medicine (www.dgklrfb.de), belonging to the program HM1 (Survey for the determination of testosterone by LC-MS/MS). The goal of this external quality assessment is to demonstrate the competence through participation in inter-laboratory trials. This material was processed and analyzed at the same time as the study samples; the results obtained did not differ by more than 10% with regard to their certified value, therefore verifying the accuracy of our method.”

Comment 3: Insufficient information is provided on the generation of the ROC data. Nothing is mentioned in the methods section as to how subjects are diagnosed by a reference method leading to the sensitivity and specificity data.

Response 3: Thanks for your remark.

Information for the generation of the ROC data, as well as, information about how subjects are diagnosed by a reference method leading to the sensitivity and specificity data has been added in the Statistical analysis section in the revised manuscript (Lines 211-216):

- “The clinical accuracy of the IA was analyzed by using Receiver Operator Characteristic (ROC) plots. The ROC area under the curve (AUC) was calculated using LC-MS/MS data as the gold standard. Sensitivity was defined as the proportion of patients correctly identified as having hypoandrogenemia, as initially diagnosed using LC-MS/MS in serum samples. Specificity was defined as the proportion of study participants accurately identified as negative for hypoandrogenemia.”

Comment 4: Measurements by LC-MS/MS should be excluded from the multivariate model. LC-MS/MS measurements are an intrinsic part of the dependent variable you are trying to explain and have no added value here.

Response 4: As indicated by the reviewer, measurements by LC-MS/MS have been excluded from the multivariate model.

Thus, we have done a new analysis of data and results have been changed in the revised version of the manuscript (Table 3; lines 281-282 and lines 343-344):

Lines 281-282

“In this multiple logistic regression analysis, the optimum model that best explained the presence of testosterone bias included the insulin resistance (p=0.026) (Table 3)”

Lines 343-344

“It is important to highlight that in our study, the only factor associated with discrepant IA values was insulin resistance, which is a common finding is patients with obesity.”

Comment 5: Often the text is difficult to read due to incorrect punctuation, missing/wrong words or confusing paragraph structures. Before resubmission, I strongly recommend the authors to either improve the text themselves or send the text to an independent text editor.

Response 5: As suggested by the reviewer, our manuscript has been submitted to full professional English editing and proofreading in order to improve the writing style and readability.

Minor criticisms

Comment 6: The authors refer to liquid chromatography tandem-mass spectrometry as LC_MS. Use either LC-MSMS or LC-MS/MS.

Response 6: As suggested by the reviewer, the acronym LC_MS had been replaced throughout the revised manuscript by LC-MS/MS.

Comment 7: The data in Figure 1 is largely already represented in Table 1. Therefore, Figure 1 has limited added value.

Response 7: Thanks for your commentary. We also agree with the reviewer that figure 1 is somehow redundant, and therefore, it has been removed.

Comment 8: The scales of Figure 2 are confusing. The x-axis starts at 1, while the y-axis starts at 0. This should be changed for the x-axis into 0.

Response 8: Thanks for your remark. As suggested, we have modified figure 2 and now x-axis starts at 0.

Round 2

Reviewer 1 Report

  1. Actually, the authors did not fully respond to the remark about the percentage of hypoandrogenism found with IA method. My main concern was relative to the fact that, in my opinion, it is not correct to classify the hypoandrogenism with IA using a cutoff of LC-MS, regardless whether the latter could be more accurate. Considering the underestimation of the absolute values of testosterone by the IA, it is obvious that this method will determine a higher number of hypoandrogenisms.               To compare the performances of the two methods the corresponding cut-off should be used. Then, it should be evaluated the possible discordances between the two methodologies.
  1. The information describing how LoQ was determined is a bit confused but it seems similar to the determination of the LoD. On the other hand, the description of the determination of the LoD actually correspond to the determination of the Limit of Blank (LoB). Please refer to the CLSI EP17-A2

Author Response

1- Actually, the authors did not fully respond to the remark about the percentage of hypoandrogenism found with IA method. My main concern was relative to the fact that, in my opinion, it is not correct to classify the hypoandrogenism with IA using a cutoff of LC-MS, regardless whether the latter could be more accurate. Considering the underestimation of the absolute values of testosterone by the IA, it is obvious that this method will determine a higher number of hypoandrogenisms. To compare the performances of the two methods the corresponding cut-off should be used. Then, it should be evaluated the possible discordances between the two methodologies.

Response:Thanks for your commentary.

We agree with the reviewer that a different cut-off for testosterone could be used depending if the testosterone determination has been done with IA or with LC-MS.

Unfortunately, there are no differentiated cut-offs for both techniques, and recommended cut-offs for total testosterone are based on LC-MS determinations, since LC-MS assays generally offer higher concentrations of specificity, sensitivity, and precision (especially in the low range) than do most immunoassays, and also, the majority of hypoandrogenemia cut-offs are based on studies that have used LC-MS.

Thus, taking into consideration the reviewer´s recommendation, we have further clarified the section “2.4 Definition of hypoandrogenemia” (lines 203-206):

- “We used testosterone values from the LC-MS/MS assay as the reference for diagnosing hypoandrogenemia, since LC-MS assays generally offers higher concentrations of specificity, sensitivity, and precision (especially in the low range) than do most immunoassays and, also, testosterone reference ranges are mostly based on LC-MS assays (8).”

 Also, we have included a new sentence in the discussion section (lines 345-349):

- “It is important to highlight that we did not use different cut-off values for hypoandrogenemia depending if the testosterone determination was done with IA or with LC-MS, and we used the LC-MS/MS values as the reference. The rationale for this was that no differentiated cut-offs values for both techniques are available and, in addition, most clinical guidelines use testosterone cut-off values derived from LC-MS/MS assays (8).”

2- The information describing how LoQ was determined is a bit confused but it seems similar to the determination of the LoD. On the other hand, the description of the determination of the LoD actually correspond to the determination of the Limit of Blank (LoB). Please refer to the CLSI EP17-A2.  

Response:Thanks for your commentary. To describe how the LoQ, LoB and LoD were determined we have now used the CLSI EP17-A2 guideline. We have added this paragraph in the revised manuscript (lines 138-144)

“Detection limit studies were carried out in accordance with the CLSI EP17-A2 guideline. For IA, the limit of blank (LoB) was determined from 20 replicates of a zero material (the manufacturer provided analyte-free diluent). Limit of detection (LoD) was determined from limit of blank and using the following equation: LoD = LoB + cβSDs where SDs is the estimated standard deviation of the sample distribution at a low level and cβ is derived from the 95th percentile of the standard Gaussian distribution. The LoQ pools were prepared by diluting patient sera. The LoQ is the lowest concentration that can be detected with ≤ 20%CV”.

Reviewer 3 Report

Major criticisms

Introduction

Academic writing is improved greatly and the added text on the method comparisons between the IA and mass spectrometry-based assays is of added value. However, I would recommend to improve the structure of the introduction somewhat before publication. The paragraph beginning with the discussion of the landmark study performed by Taieb et al. is the beginning of a very/too long continuous text ending with a description of LC-MS/MS in routine diagnostics. Subsequently, yet another IA/LC-MS/MS comparison is highlighted. The authors should consider restructuring the introduction to improve reading.

Materials and methods

Interference:

Blanks were measured to detect interference and if an interfering signal was detected it was subtracted from sample concentrations. This is an unsatisfactory method to deal with interferences. Firstly, blanks cannot be generalized to the rest of the samples. It could be, for instance, that the blank sample suffers from carry-over, of which the effect is likely to diminish in the consecutive samples. Therefore, subtracting signal or concentration from other samples would potentially result in falsely quantitated samples. Secondly, it is standard operating procedure to perform extensive interference testing of 1) serum parameters (lipemia, icterus, hemolysis), 2) structural analogs (similarly structured steroids) and 3) frequently prescribed medication in your target population. These compounds should not interfere with the quantitation of your analyte, i.e. testosterone.

ROC analysis:

In this analysis, the IA is tested for its ability to diagnose hypoandrogenemia with the LC-MS method as the reference method. However, the LC-MS assay used for this analysis is still under peer review and not a method from a reference lab. Therefore, it can be questioned whether this is an appropriate reference method for this analysis.

Results

Patient characterization

In addition to the median values, the author has included the P 10 and P 90 values. Assuming these represent the 10th and 90th percentiles (no abbreviation listed), it is inappropriate to provide an 80% interval. Using median values, either provide 2.5th and 97.5th percentiles (95%confidence interval) or the whole range.

Logistic regression

In the logistic regression, factors that potentially explain a >20% lower IA value compared to LC-MS are investigated. Looking at the PB regression and the BA bias plot it is clear that there is a relative bias of approximately 20% between the methods (i.e. slope CI does not include 1). These biases are mostly explained by differences in method standardization/calibration. Another dependent variable would therefore be more suitable, such as biases that differ >20% from the mean bias.

Minor criticisms

Material and methods

127-128: Beginning with “All samples”. Sentence is unclear. At each test facility samples were measured similarly?

On SPE. What washing/elution solution was used?

Samples were reconstituted in what solvent?

Chromatographic method was not provided.

Which mobile phases were used.

Results

In table 3 the 95%CI of HOMA-IR > 1.85 does not include the estimated OR.

Author Response

1- Academic writing is improved greatly and the added text on the method comparisons between the IA and mass spectrometry-based assays is of added value. However, I would recommend to improve the structure of the introduction somewhat before publication. The paragraph beginning with the discussion of the landmark study performed by Taieb et al. is the beginning of a very/too long continuous text ending with a description of LC-MS/MS in routine diagnostics. Subsequently, yet another IA/LC-MS/MS comparison is highlighted. The authors should consider restructuring the introduction to improve reading. 

Response:Thanks for your pertinent commentary. As indicated by the reviewer we have restructured the introduction to improve reading (Lines 65-88).  

Materials and methods

2- Interference:

Blanks were measured to detect interference and if an interfering signal was detected it was subtracted from sample concentrations. This is an unsatisfactory method to deal with interferences. Firstly, blanks cannot be generalized to the rest of the samples. It could be, for instance, that the blank sample suffers from carry-over, of which the effect is likely to diminish in the consecutive samples. Therefore, subtracting signal or concentration from other samples would potentially result in falsely quantitated samples. Secondly, it is standard operating procedure to perform extensive interference testing of 1) serum parameters (lipemia, icterus, hemolysis), 2) structural analogs (similarly structured steroids) and 3) frequently prescribed medication in your target population. These compounds should not interfere with the quantitation of your analyte, i.e. testosterone.

Response: Thanks for your commentary. We are sorry about the mistake in the description of the concept interference. We have added a new paragraph about the elimination of interferences in our LC-MS/MS method (lines 189-192):

 “Elimination of interferences in the measurement system can be achieved using different MRM channels for the same compound. Multiple reaction monitoring also enables application of the isotopically labelled internal standards (ISTD), which improves the accuracy”

3- ROC analysis:

In this analysis, the IA is tested for its ability to diagnose hypoandrogenemia with the LC-MS method as the reference method. However, the LC-MS assay used for this analysis is still under peer review and not a method from a reference lab. Therefore, it can be questioned whether this is an appropriate reference method for this analysis.

Response:Thanks for your commentary. We have explained in the section 2.3.2. LC-MS/MS, that the LC-MS/MS method was verified using validated external quality controls (lines 195-201):

“To ensure the quality of the LC-MS/MS results, external controls (serum samples) were used as a reference. These quality controls were obtained from the German Society for Clinical Chemistry and Laboratory Medicine (www.dgklrfb.de), belonging to the program HM1 (Survey for the determination of testosterone by LC-MS/MS). The goal of this external quality assessment is to demonstrate the competence through participation in inter-laboratory trials. This material was processed and analyzed at the same time as the study samples; the results obtained did not differ by more than 10% with regard to their certified value, therefore verifying the accuracy of our method.”

Results

Patient characterization

In addition to the median values, the author has included the P 10 and P 90 values. Assuming these represent the 10th and 90th percentiles (no abbreviation listed), it is inappropriate to provide an 80% interval. Using median values, either provide 2.5th and 97.5th percentiles (95%confidence interval) or the whole range.

Response:Thanks for your remark. We have provided the 2.5 and 97.5 percentiles (95% confidence interval) for the median value in the table 1 in the revised manuscript.

We included the 10 and 90 percentiles in the previous version of the manuscript as was suggested by another reviewer, who considered that mean and SD were not enough to characterize the population and asked us to provide median and percentiles. We are happy to include in this new version of the manuscript percentiles 2.5 and 97.5.

Logistic regression

In the logistic regression, factors that potentially explain a >20% lower IA value compared to LC-MS are investigated. Looking at the PB regression and the BA bias plot it is clear that there is a relative bias of approximately 20% between the methods (i.e. slope CI does not include 1). These biases are mostly explained by differences in method standardization/calibration. Another dependent variable would therefore be more suitable, such as biases that differ >20% from the mean bias.

Response:We appreciate the observation of the reviewer.

Following the reviewer´s commentary, we have performed a new logistic regression analysis, including biases that differ >20% from the mean bias as the dependent variable.

Please, take into account that the results have slightly changed, and in the new version of the manuscript, SHBG concentrations have become another factor associated with IA discrepancy.

Thus, we have modified lines 296-306:

- “To evaluate the factors associated with the systematic IA bias in the determination of total testosterone, a multivariate model was constructed, considering as the dependent variable the biases between IA and LC_MS/MS determinations that differed >20% from the mean bias. In this multiple logistic regression analysis, the optimum model that best explained the presence of testosterone bias included SHBG concentrations (p=0.002) and insulin resistance (p=0.033) (Table 3).”

Also, we have modified table 3, and we have changed the discussion section (lines 366-369):

“Finally, in our study, we found that the factor associated with discrepant IA values were SHBG concentrations and insulin resistance. These findings are in agreement with the fact that insulin resistance is a common finding in patients with obesity and with the close interrelation between obesity and SHBG homeostasis (7).”

Minor criticisms

Material and methods

127-128: Beginning with “All samples”. Sentence is unclear. At each test facility samples were measured similarly? 

Response:Thanks for your commentary.We have included a new sentence to clarify this point (lines 115-118):

- “All samples were collected in the Outpatient Clinic of the Endocrinology Department of the Virgen de la Victoria Hospital, and were immediately transported to the Medical Laboratory Department where they were bar-coded and blinded to patient identification”.

On SPE. What washing/elution solution was used? Samples were reconstituted in what solvent? 

Response:The washing/ elution solution and solvent for sample reconstitution were provided by the MassChrom Steroids kit (Chromsystems, Munich, Germany) used for TT determination. 

We have added this new information in the section 2.3.2 LC-MS/MS (M&M section) (lines 151-154). 

“TT determination was carried out using MassChrom Steroids kit (Chromsystems, Munich, Germany) and high performance LC-MS/MS triple quadrupole equipment (Model 6460, Agilent Technologies). The applied conditions for sample treatment and MS/MS determination were the indicated by the kit with slight modifications”.

Chromatographic method was not provided. Which mobile phases were used.

Response: Thanks for your remark. This information has been added in the section 2.3.2 LC-MS/MS (M&M section) (lines 159-167) and (lines 168-176).   

“Two hundred microliters of calibrators, QC, and sample, along with 50 ul of the internal standard (testosterone-d3), were subjected to a process of reversed phase 96-well solid-phase extraction prior to injection into the HPLC system. The eluates were evaporated to dryness by means of a nitrogen stream at 45°C and reconstituted in 100 ul of initial mobile phase. To carry out the analysis, 20 μl of processed sample were injected into the chromatographic system and were separated onto an HPLC column (Chromsystems, Gräfelfing, Germany). Two MRM transitions for testosterone were measured to check for interference (289.2─97 m/z and 289. 2─109 m/z) and one transition for testosterone-d3 (292.2─97 m/z)”.

Lines 168-176

 “The gradient used for TT quantification was 70% of phase A (aqueous phase with methanol) and 30% of phase B (water /acetonitrile) with a flow of 0.8 mL / min during the first minute of the analysis. Then, the percentage of phase B increased from 30% to 44% and continued until 3.50 min where the flow was lowered to 0.4 mL / min. This flow was maintained until min 7, where the percentage of phase B increased to 50% and a flow of 0.6 mL / min. In min 8.61 the flow rose to 0.8 mL / min and the percentage of phase B to 60%, reaching 100% and a flow of 1 mL / min in min 10.81. From min 11.51 they returned to the initial conditions until min 13. Finally, 2 min were left under these conditions to re-equilibrate the column until next injection”.

Results

In table 3 the 95% CI of HOMA-IR > 1.85 does not include the estimated OR. 

Response:Thanks for your remark. It has been modified in the new version of table 3.